# Dirichlet-based Uncertainty Quantification for Personalized Federated Learning with Improved Posterior Networks

## Abstract

In modern federated learning, one of the main challenges is to account for inherent heterogeneity and the diverse nature of data distributions for different clients. This problem is often addressed by introducing personalization of the models towards the data distribution of the particular client. However, a personalized model might be unreliable when applied to the data that is not typical for this client. Eventually, it may perform worse for these data than the non-personalized global model trained in a federated way on the data from all the clients. This paper presents a new approach to federated learning that allows selecting a model from global and personalized ones that would perform better for a particular input point. It is achieved through a careful modeling of predictive uncertainties that helps to detect local and global in- and out-of-distribution data and use this information to select the model that is confident in a prediction. The comprehensive experimental evaluation on the popular real-world image datasets shows the superior performance of the model in the presence of out-of-distribution data while performing on par with state-of-the-art personalized federated learning algorithms in the standard scenarios.

## 1 Introduction

The widespread adoption of deep neural networks in various applications requires reliable predictions, which can be achieved through rigorous uncertainty quantification. Although uncertainty quantification has been extensively studied in different domains under centralized settings (Lee et al., 2018; Lakshminarayanan et al., 2017; Gal & Ghahramani, 2016), only a few works have considered this area within the context of federated learning (Linsner et al., 2022; Kotelevskii et al., 2022). Typically, in federated learning papers, algorithms result in using either a personalized local model or a global model. However, both these models could be useful in different cases by providing the tradeoff between personalization of a local model and higher reliability of the global one (Hanzely & Richtárik, 2020; Liang et al., 2019).

In this paper, we introduce a new framework to choose whether to predict with a local or global model at a given point based on uncertainty quantification. The core idea is to apply the global model only if the local one has high epistemic (model) uncertainty (Hüllermeier & Waegeman, 2021) about the prediction at a given point, i.e., the local model doesn't have enough information about the particular input point. In case the local model is confident (either in predicting a particular class or in the fact that it is observing an ambiguous object with high aleatoric (data) uncertainty (Hüllermeier & Waegeman, 2021)), it should make the decision itself without involving the global one.

As a specific instance of our framework, we propose an approach inspired by Posterior Networks (`PostNet`; Charpentier et al. (2020)) and its modification, Natural Posterior Networks (`NatPN`; Charpentier et al. (2022)). This type of model is particularly useful for our purposes, as it enables the estimation of aleatoric and epistemic uncertainties without incurring additional inference costs. Thus, we are able to fully implement the switching between local and global models in an efficient way.

**Related work.** The literature presents various approaches to uncertainty quantification in federated learning settings. In (Linsner et al., 2022), the authors suggest that training an ensemble of $K$ global models is the most effective for federated uncertainty quantification. Despite its effectiveness, this approach is $K$ times more expensive compared to the classic `FedAvg` method. Another proposal comes from (Kotelevskii et al., 2022), where the authors recommend using Markov Chain Monte

Carlo (MCMC) to obtain samples from the posterior distribution. However, this method is practically almost infeasible due to its significant computational complexity.

Other works, such as (Chen & Chao, 2021; Kim & Hospedales, 2023), also present methods that could potentially estimate uncertainty in federated learning. However, these papers do not expressly address the opportunities and challenges of uncertainty quantification in their discussion. It's important to note that there are existing approaches of deferring classification to other models or experts in case of abstention, for example (Keswani et al., 2021). Despite this, none of these approaches have been explored in a federated context, nor have they considered the corresponding constraints.

The central idea of `PostNet` and `NatPN` involves using a Dirichlet prior and posterior distributions over the categorical predictive distributions (Malinin & Gales, 2018; 2019; Charpentier et al., 2020; 2022; Sensoy et al., 2018). To parameterize the parameters of these Dirichlet distributions, the authors suggest introducing a density model over the deep representations of input objects. In `NatPN`, a Normalizing Flow (Papamakarios et al., 2021; Kobyzev et al., 2020) is employed to estimate the density of embeddings extracted by a trained feature extractor. This density is then used to calculate updates to the Dirichlet distribution.

Despite the success of the `NatPN` model (Charpentier et al., 2022), we identified certain issues with the loss function employed in `NatPN`, which become particularly critical when dealing with high aleatoric uncertainty regions. Intriguingly, the issue was not discovered in the authors' recent work (Charpentier et al., 2023), which investigated potential issues in the training procedure. Recent works (Bengs et al., 2022; 2023) have unveiled other potential issues related to the training of Dirichlet models in general but have not offered solutions to address these challenges.

**Contributions.** Our paper stands out as one of the few studies that specifically addresses the problem of uncertainty quantification in federated learning. We aim to provide a comprehensive analysis of this important topic, considering both theoretical and practical aspects, and presenting solutions that overcome the limitations of existing approaches in the literature. The contributions of this paper are as follows:

1. We present a new federated learning framework that is based on uncertainty quantification, which allows *switching* between using a personalized local or global model. To achieve that, we consider several types of uncertainties for local and global models and propose a procedure that ensures that the model is confident in its prediction if the prediction is made. Otherwise, our framework rejects the prediction if we are confident that there is a high uncertainty in the predicted class.
2. We introduce a specific realization of our framework `FedPN`, using Dirichlet-based `NatPN` model. For this particular model, we identify an issue in the loss function of `NatPN` (not known in literature before) that complicates disentanglement of aleatoric and epistemic uncertainties, and propose a solution to rectify it.
3. We conduct an extensive set of experiments that demonstrate the benefits of our approach for different input data scenarios. In particular, we show that the proposed model outperforms state-of-the-art personalized federated learning algorithms in the presence of out-of-distribution date, while being on par with them in standard scenarios.

## 2 GENERAL FRAMEWORK OF SWITCHING BETWEEN GLOBAL AND PERSONAL MODELS

In this section, we introduce our framework, discussing the general idea and potential nuances. In the subsequent section, we will delve into a specific implementation.

### 2.1 CONCEPT OVERVIEW

We are going to consider a federated learning setting with multiple clients, each having its own personalized local model. However, we additionally assume that the global model is available. The global model is typically expected to perform reasonably well on each client's data. Assuming that we already have trained global and local models, we consider a situation where clients have the option to use either the global model or their local models to make predictions for a new incoming object $x$.

The choice between local and global models for prediction depends on the multiple factors that contribute to their prediction quality. First of all, shift in the distribution between the local data of a particular client and the global population may have significant effect on the models' performance.

Possible shifts include covariate shift, label shift or different types of label noise. If the shift is significant, the global model might be very biased with respect to the prediction for the particular client, while normally the local model is unbiased.

The second part of the picture is the size of the available data. Generally, the global model has more data and potentially, if no data shift is present, should outperform the local one. However, the global model is usually trained with no direct access to the data stored at clients, which might degrade its performance. Eventually, the best performing model will be the one which achieves better bias variance trade-off.

In this work, we propose a framework to choose between the pointwise usage of a local or global model for prediction. This decision is based on the uncertainty scores provided by the model. This approach aims to mitigate issues that arise when there is a distribution shift between a client's distribution and the global distribution, which can often hinder model performance. By using uncertainty to guide the model selection, we can enhance the model's ability to perform well even in the presence of such distribution shifts.

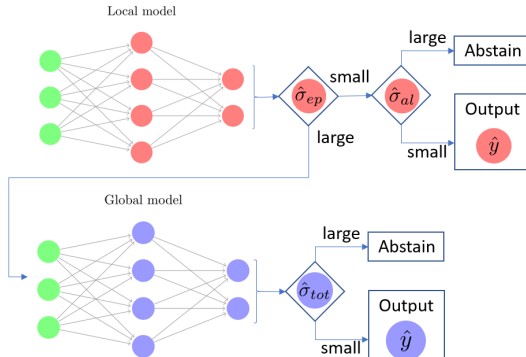

Figure 1: The general scheme of the proposed approach. Each input is first processed by personalized local model. In case of high epistemic uncertainty, the decision is delegated to the global model. Otherwise, if epistemic uncertainty is low, local model proceeds with the decision. Both models consider also aleatoric uncertainty and may abstain from prediction.

Both the local and global models can provide uncertainty estimates. These estimates include not only the total predictive uncertainty but also separately aleatoric and epistemic uncertainties. This feature allows for a more refined decision-making process, ensuring the most suitable model is used for prediction or choosing to abstain from making a prediction altogether in the face of high uncertainty. The resulting workflow is summarized in Figure 1.

## 2.2 OPTIMAL CHOICE OF MODEL

We start from the important fact that the local model is not exposed to the data shift between the general population and the particular client. Thus, if this model is sufficiently confident in the prediction, there is no need to involve the global model at all. However, it is important to distinguish between different types of uncertainty here. Usually, the total uncertainty of the model predicting at a particular data point can be split into two parts: aleatoric uncertainty and epistemic uncertainty (Hüllermeier & Waegeman, 2021). *Aleatoric* uncertainty is the one that reflects the inherent randomness in the data labels. The *epistemic* uncertainty is the one that reflects the lack of knowledge due to the fact that the model was trained on a data set of a limited size, or model misspecification. In what follows we will refer to epistemic as to "lack of knowledge" uncertainty, as the models we are using, neural networks, are known to be very flexible.

For our problem, it is extremely important to distinguish between aleatoric and epistemic uncertainties. Suppose the local model has low epistemic and high aleatoric uncertainty at some point. In that case, the model is confident that the predicted label is ambiguous, and the model should abstain from prediction. However, if the epistemic uncertainty is high, it means that the model doesn't have enough knowledge to make the prediction (not enough data), and the global model should make the decision. The global model, in its turn, may either proceed with the prediction if it is confident or abstain from prediction if there is high uncertainty associated with the prediction. Thus, in this context, for a fixed client and an unseen input, there are four interesting outcomes, see Table 1. Note, that when local epistemic uncertainty is low, there is no need to refer to the global model. The inherent assumption here is that the local model is better than the global one if it knows the input point well. Thus we consider only two options for the local model: low epistemic with low aleatoric and low epistemic with high aleatoric regardless of the confidence of the global model. When the local epistemic uncertainty is high, it means that the client barely knows the input point, and thus we refer to the global model. For the global model, we look at the total uncertainty as we only care about the error of prediction which is determined by total uncertainty.

The particular implementation of the approach described above would depend on the choice of the

| Known knowns | Known unknowns |
|---|---|
| **Local Confident.** This represents local in-distribution data for which the local model is confident in prediction. | **Local Ambiguous.** This is local in-distribution data with high aleatoric uncertainty (class ambiguity). |
| Unknown knowns | Unknown unknowns |
| **Local OOD.** This refers to data that is locally unknown (high epistemic uncertainty) but known to other clients. In this case, it makes sense to use the global model for predictions. | **Global Uncertain.** These input data is out-of-distribution for the local model while the global model is uncertain in prediction (high total uncertainty). The best course of action is to abstain from making a prediction. |

Table 1: Possible scenarios for the input data point in the introduced setup. A particular input falls in one of the categories depending on the confidence in its prediction by local and global models. The disentanglement between aleatoric and epistemic uncertainties is crucial to make the decision in an optimal way.

machine learning model and the way to compute uncertainty estimates. The key feature required is the ability of the method to compute both aleatoric and epistemic uncertainties. In the next section, we propose the implementation of the method based on the posterior networks framework (Charpentier et al., 2020).

## 3 DIRICHLET-BASED DEEP LEARNING MODELS

We have chosen to showcase Dirichlet-based models (Malinin & Gales, 2018; Charpentier et al., 2020; 2022) as a specific instance of our general framework. The intuition behind this decision lies in the fact that these models allow the distinction between various types of uncertainty and facilitate the computation of corresponding uncertainty estimates with minimal additional computational overhead. Furthermore, unlike ensemble methods (Beluch et al., 2018), there is no need to train multiple models. In comparison to approximate Bayesian techniques, such as MC Dropout (Gal & Ghahramani, 2016), Variational Inference (Graves, 2011) or MCMC (Izmailov et al., 2021), almost all expectations of interest can be derived in closed form and almost without computational overhead. This makes Dirichlet-based models an attractive and efficient option for implementing our proposed framework.

### 3.1 BASICS OF DIRICHLET-BASED MODELS

In this section, we introduce the basics of Dirichlet-based models for classification tasks. To ease the introduction, let us start by considering a training dataset $D = \{x_i, y_i\}_{i=1}^{N}$, where $N$ denotes the total number of data points in the dataset. We assume that labels $y_i$ belong to one of the $K$ classes.

Typically, the Dirichlet-based approaches assume that the model consists of two hierarchical random variables, $\boldsymbol{\mu}$ and $\theta$. The posterior predictive distribution for a given unseen object $x$ can be computed as follows:

$$p(y \mid x, D) = \int p(y \mid \boldsymbol{\mu}) \left[ \int p(\boldsymbol{\mu} \mid x, \theta)\, p(\theta \mid D) d\theta \right] d\boldsymbol{\mu},$$

where $p(y \mid \boldsymbol{\mu})$ is the distribution over class labels, given some probability vector (e.g., Categorical), $p(\boldsymbol{\mu} \mid x, \theta)$ is the distribution over a simplex (e.g., Dirichlet), and $p(\theta \mid D)$ is the posterior distribution over parameters of the model.

However, for practical neural networks, the posterior distribution $p(\theta \mid D)$ does not have an analytical form and is computationally intractable. Following (Malinin & Gales, 2018), we suggest considering "semi-Bayesian" scenario by looking on a point estimate of this distribution: $p(\theta \mid D) = \delta(\theta - \hat{\theta})$, where $\hat{\theta}$ is some estimate of the parameters (e.g., MAP estimate). Then the integral inside the brackets simplifies:

$$\int p(\boldsymbol{\mu} \mid x, \theta)\, p(\theta \mid D) d\theta = \int p(\boldsymbol{\mu} \mid x, \theta)\, \delta(\theta - \hat{\theta}) d\theta = p(\boldsymbol{\mu} \mid x, \hat{\theta}).$$

In the series of works (Malinin & Gales, 2018; 2019; Charpentier et al., 2020; 2022) the posterior distribution $p(\boldsymbol{\mu} \mid x, \hat{\theta})$ is chosen to be the Dirichlet distribution $Dir(\boldsymbol{\mu} \mid \boldsymbol{\alpha}^{\text{post}}(x))$ with the parameter vector $\boldsymbol{\alpha}^{\text{post}}(x) = \boldsymbol{\alpha}^{\text{post}}(x \mid \hat{\theta})$ that depends on the input point $x$. In these models, the prior

over probability vectors $\boldsymbol{\mu}$ takes the form of a Dirichlet distribution, representing the distribution over our beliefs about the probability of each class label. In other words, it is a distribution over distributions of class labels. This prior is parameterized by a parameter vector $\boldsymbol{\alpha}^{\text{prior}}$, where each component $\alpha_c^{\text{prior}}$ corresponds to our belief in a specific class. `PostNet` (Charpentier et al., 2020) and `NatPN` (Charpentier et al., 2022) propose the idea that the posterior parameters $\boldsymbol{\alpha}^{\text{post}}(x)$ can be computed in the form of pseudo-counts that are computed by a function:

$$\boldsymbol{\alpha}^{\text{post}}(x) = \boldsymbol{\alpha}^{\text{prior}} + \boldsymbol{\alpha}(x), \tag{1}$$

where $\boldsymbol{\alpha}(x) = \boldsymbol{\alpha}(x \mid \hat{\theta})$ is a function of input object $x$ that maps it to positive values.

**Parameterization of $\boldsymbol{\alpha}(x)$.** In `NatPN` it is proposed to use the following parameterization:

$$\boldsymbol{\alpha}(x) = p\big(g(x)\big)\boldsymbol{f}\big(g(x)\big). \tag{2}$$

In this parameterization, $g(x)$ represents a feature extraction function that maps the input object $x$ (usually high-dimensional) to a lower-dimensional embedding. Subsequently, $p(\cdot)$ is a "density" function (parameterized by normalizing flow in the case of (Charpentier et al., 2022; 2020)), and $\boldsymbol{f}(\cdot)$ is a function mapping the extracted features to a vector of class probabilities.

This parameterization offers several advantages. Firstly, since $p(\cdot)$ is expected to represent the density of training examples, it should be high for in-distribution data. Secondly, as the density is properly normalized, embeddings that lie far from the training ones will result in lower values of $p\big(g(x)\big)$, thus leading to lower $\boldsymbol{\alpha}(x)$. This means that for such input $x$, we will not add any evidence, and consequently, $\boldsymbol{\alpha}^{\text{post}}(x)$ will be close to $\boldsymbol{\alpha}^{\text{prior}}$.

### 3.2 UNCERTAINTY MEASURES FOR DIRICHLET-BASED MODELS

One of the advantages of using Dirichlet-based models is their ability to easily disentangle and quantitatively estimate aleatoric and epistemic uncertainties.

**Epistemic uncertainty.** We begin by discussing epistemic uncertainty, which can be estimated in multiple different ways (Malinin & Gales, 2021). In this work, following the ideas from (Malinin & Gales, 2018), we quantify the epistemic uncertainty as the entropy of a posterior Dirichlet distribution, which can be analytically computed as follows

$$\mathcal{H}[Dir\big(\boldsymbol{\mu} \mid \boldsymbol{\alpha}^{\text{post}}(x)\big)] = \ln \frac{\prod_{i=1}^{K} \Gamma\big(\alpha_i^{\text{post}}(x)\big)}{\Gamma\big(\alpha_0^{\text{post}}(x)\big)} - \sum_{i=1}^{K}(\alpha_i^{\text{post}}(x)-1)(\tilde{\psi}\big(\alpha_i^{\text{post}}(x)\big)-\tilde{\psi}\big(\alpha_0^{\text{post}}(x)\big)), \tag{3}$$

where $\boldsymbol{\alpha}^{\text{post}}(x) = \big[\alpha_1^{\text{post}}(x), \ldots, \alpha_K^{\text{post}}(x)\big]$, $\tilde{\psi}$ is a digamma function and $\alpha_0^{\text{post}}(x) = \sum_{i=1}^{K} \alpha_i^{\text{post}}(x)$.

**Aleatoric uncertainty.** Aleatoric uncertainty can be measured using the average entropy (Kendall & Gal, 2017), which can be computed as follows:

$$\mathbb{E}_{\boldsymbol{\mu} \sim Dir\big(\boldsymbol{\mu}|\boldsymbol{\alpha}^{\text{post}}(x)\big)}\mathcal{H}[p(y \mid \boldsymbol{\mu})] = -\sum_{i=1}^{K} \frac{\alpha_i^{\text{post}}(x)}{\alpha_0^{\text{post}}(x)}[\tilde{\psi}\big(\alpha_i^{\text{post}}(x) + 1\big) - \tilde{\psi}\big(\alpha_0^{\text{post}}(x) + 1\big)]. \tag{4}$$

This metric captures the inherent noise present in the data, thus providing an estimate of the aleatoric uncertainty.

### 3.3 LOSS FUNCTIONS FOR DIRICHLET-BASED MODELS

The loss function used in (Charpentier et al., 2022; 2020) is the expected cross-entropy, also known as Uncertain Cross Entropy (UCE) (Biloš et al., 2019). Note, that this loss function is a *strictly proper scoring rule*, which implies that a learner is incentivized to learn the true conditional $p(y \mid x)$. For a given input $x$, the loss can be written as:

$$L(y, \boldsymbol{\alpha}^{\text{post}}(x)) = \mathbb{E}_{\boldsymbol{\mu} \sim Dir\big(\boldsymbol{\mu}|\boldsymbol{\alpha}^{\text{post}}(x)\big)}\sum_{i=1}^{K} - \mathbb{1}[y = i] \log \mu_i =$$

$$\mathbb{E}_{\boldsymbol{\mu} \sim Dir\big(\boldsymbol{\mu}|\boldsymbol{\alpha}^{\text{post}}(x)\big)}\text{CE}(\boldsymbol{\mu}, y) = \tilde{\psi}\big(\alpha_0^{\text{post}}(x)\big) - \tilde{\psi}\big(\alpha_y^{\text{post}}(x)\big), \tag{5}$$

where $\alpha_0^{\text{post}}(x) = \sum_{i=1}^{K} \alpha_i^{\text{post}}(x)$. Additionally, authors suggest to penalize too concentrated predictions, by adding the regularization term with some hyperparameter $\lambda$. The overall loss function looks as follows:

$$L(y, \boldsymbol{\alpha}^{\text{post}}(x)) - \lambda\mathcal{H}[Dir\big(\boldsymbol{\mu} \mid \boldsymbol{\alpha}^{\text{post}}(x)\big)], \tag{6}$$

where $\mathcal{H}$ denotes the entropy of a distribution. This overall loss function referred by authors as Bayesian loss (Charpentier et al., 2022; 2020).

**Issue with the loss function.** Let us explore an asymptotic form of (5). For all $x > 0$, the following

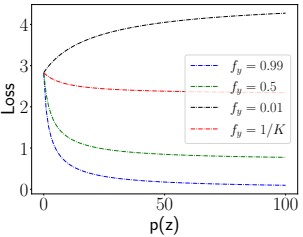 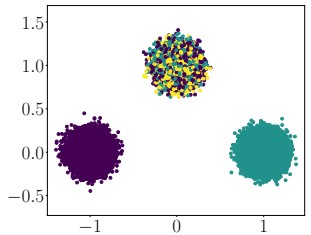 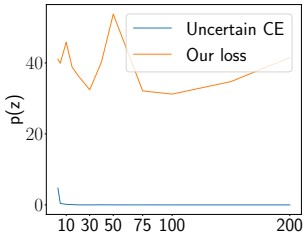

Figure 2: **Left:** Image depicts the landscape of the loss function with $K = 10$ classes. The red line (representing high aleatoric regions) appears relatively flat. Even at this level, the density in corresponding points tends to be underestimated. As the number of classes increases, this effect is amplified. **Center:** This image depicts the training data, where the leftmost and rightmost Gaussians consist of only one class each. In contrast, the middle Gaussian contains all $K$ labels, randomly permuted. Each Gaussian has an equal number of data points, implying that, if $p(z)$ follows the intuition of density, all peaks should have the same height. **Right:** By altering the number of classes $K$, we change the aleatoric uncertainty. We train the model using the vanilla approach of `NatPN` and measure $p(z)$ at the central peak. As the number of classes increases, the density for UCE decreases, violating the intuition that $p(z)$ is the density of training data. While for our proposed trick it behaves as expected.

inequality holds:

$$\log x - \tfrac{1}{x} \leq \tilde{\psi}(x) \leq \log x - \tfrac{1}{2x}.$$

Recalling the update rule (1) and using the specific parameterization of $\alpha_c^{\text{prior}} = 1$ for all $c$, we conclude that all $\alpha_c^{\text{post}}(x) > 1$. Hence, we can approximate $\tilde{\psi}\big(\alpha_c^{\text{post}}(x)\big) \approx \log \alpha_c^{\text{post}}(x)$.

To simplify the notation, we will use $z = g(x)$ and denote by $f_y(z)$ predicted probability for the $y$-th (correct) class:

$$L(y, \boldsymbol{\alpha}^{\text{post}}(x)) \approx \log(\alpha_0^{\text{post}}(x)) - \log(\alpha_y^{\text{post}}(x)) = \log K + \log \Big[1 + \tfrac{p(z)(\tfrac{1}{K} - f_y(z))}{\alpha_y^{\text{prior}} + p(z)f_y(z)}\Big], \qquad (7)$$

see the full derivation in Supplementary Material (SM), Section A.2.

We observe from (7) that for an in-distribution case with high aleatoric uncertainty (when all classes are confused and equally probable), the last term is canceled. Note, that the presence of entropy term (that is used in (Charpentier et al., 2022; 2020) resulting to the final loss of equation (6)), which incentivizes learner to produce smooth prediction will only amplify the effect. This implies that no gradients concerning the parameters of the density model will be propagated. As a result, $p(\cdot)$, which is the density in the embedding space, may disregard regions in the embedding space that correspond to areas with a high concentration of ambiguous training examples. This *violates the intuition of $p(\cdot)$ as a data density*. Thus, uncertainty estimates based on $p(\cdot)$ cannot be used to measure epistemic uncertainty, as it ignores the regions with high aleatoric uncertainty.

In addition to the problem of confusing high-aleatoric and high-epistemic regions, we discovered that the proposed loss function defy intuition when confronted with "outliers." We define an "outlier" as an object with a predicted probability of the correct class is less than $\tfrac{1}{K}$. From (7), we observe that the last term changes its sign precisely at the point where $f_y(z) = \tfrac{1}{K}$. This implies that to minimize the loss function, we must decrease $p(z)$ at these points, which seems counterintuitive since these $z$ values correspond to objects from our training data.

Although the issue concerning equation (5) arises primarily in the asymptotic context, we can readily illustrate it through the loss function profiles (see Figure 2-left) for a range of fixed correct class prediction probabilities.

To further emphasize the problem, we examine the loss function landscape and provide a demonstrative example. Consider three two-dimensional Gaussian distributions, each with a standard deviation of 0.1 and centers at $(-1, 0), (1, 0)$, and $0, 1)$ (see Figure 2-center). Left and right clusters are set to include objects of only one class, while the middle distribution contains uniformly distributed labels, representing a high aleatoric region. Each Gaussian contains an equal amount of data. We train the `NatPN` model using a centralized approach, employing both the loss function from equation (5)

and the loss function with out fix; see equation (8). Subsequently, we evaluate the quality of the learned model by plotting the density in the center of middle Gaussian. Ideally, we would expect the density to be equal for different number of classes $K$. However, we observe the different picture with "density" estimate at the central cluster decreasing when one increases the number of clusters, see Figure 2-right. In this Figure, we plot the median of the estimated density for different loss functions. We see, that our simple fix rectifies the behaviour of the loss function.

It is essential to emphasize that addressing these issues is critical for our framework, as we need to accurately differentiate between aleatoric and epistemic uncertainties in order to select the appropriate model for a given situation. Consequently, in the following section, we propose a simple but efficient technique to rectify the aforementioned problem with the loss function, ensuring that our framework effectively distinguishes between the different types of uncertainties and makes informed model choices.

**Proposed solution.** We propose to still use parametric model to estimate density, but now our goal is to ensure that $p(\cdot)$ accurately represents the density of our training embeddings. To achieve this, we propose maximizing the likelihood of the embeddings explicitly by incorporating a corresponding term into the loss function. Simultaneously, we aim to prevent any potential impact of the Bayesian loss on the density estimation parameters, maintaining their independence.

Thus, we suggest the following loss function:
$$L\big(y, \text{StopGrad}_{p(g(x))}\,\boldsymbol{\alpha}^{\text{post}}(x)\big) - \lambda \mathcal{H}\big[Dir\big(\boldsymbol{\mu} \mid \boldsymbol{\alpha}^{\text{post}}(x)\big)\big] - \gamma \log p\big(g(x)\big), \qquad (8)$$
where $\lambda, \gamma > 0$ are hyperparameters, and $\text{StopGrad}_{p(g(x))}$ means that the gradient will be not propagated to the parameters of a density model, which parameterizes $p\big(g(x)\big)$.

We should note that while the corrected loss function is indeed somewhat ad hoc, it is computationally simple and very straightforward to implement, which is very important for modern deep learning. Compared to the original loss function, our rectified loss does not introduce any additional computational overhead – in both cases (for the old loss and the new one) the gradient will be computed only once with respect to each of the parameters.

## 4 FedPN: Specific instance of the framework

### 4.1 Federated setup

In this section we show, how one can adapt `NatPN` for federated learning.

Suppose we are given an array of datasets $D_i$ for $1 \leq i \leq b$, where $b$ represents the number of clients. Each $D_i = \{x_j^i \in \mathbb{R}^d, y_j^i\}_{j=1}^{|D_i|}$ consists of object-label pairs. We construct our federated framework in such a way that all clients share the feature extractor $g$ parameters $\phi$ and maintain personalized heads $f^i$ parameterized by $\theta_i$. Furthermore, we retain a "global" head-model $f$, which is trained using the `FedAvg` (McMahan et al., 2017) method and which ultimately has parameters $\theta$.

Following the `NatPN` approach, we employ normalizing flows to estimate the embedding density. As it is for head models, here we also learn two types of models – a local density models $p^i$ (using local data) parameterized by $\psi_i$ and global density model $p$ (using data from other clients in a federated fashion) parameterized by $\psi$. It is important to note that both types of models are trained on the same domain since the feature extractor model is fixed for local and global models. We refer readers to Appendix, Section B.2 for the discussion of computational overhead.

### 4.2 Threshold selection

Before discussing the results, it is essential to understand how we determine whether to make predictions using a local model or a global one.

This decision, resulting in a "switching" model, depends on a particular uncertainty score. This score can either be the logarithm of the density of embeddings, obtained using the density model (normalizing flows), or the entropy of predictive Dirichlet distribution (3). We found that both measures provide comparable behaviour, and in the experiments for the Table 2 we use density of embeddings.

To apply this approach, we must establish a rule for how a client decides whether to use its local model for predictions on a previously unseen input object $x$ or to delegate the prediction to the global

model. One approach is to select some uncertainty values' threshold. This threshold can be chosen based on an additional calibration dataset. In our experiments, we split each client's validation dataset in a 40/60 ratio, using the smaller part for calibration. Note that the calibration dataset only includes those classes used during the training procedure.

The choice of the threshold is arguably the most subjective part of the approach. Ideally, we would desire to have access to explicit out-of-distribution data (either from other client "local OOD" or completely unrelated data "global OOD"). With this data, we could explicitly compute uncertainty scores for both types of data (in-distribution and out-of-distribution) and select the threshold that maximizes accuracy. However, we believe it is unfair to assume that we have this data in the problem statement. Therefore, we suggest a procedure for choosing the threshold based solely on available local data.

To choose the threshold, we assume that for all clients, there might be a chance that some $p\%$ (typically 10%) of objects considered as outliers. We further compute the estimates of epistemic uncertainty (with either entropy or the logarithms of density of embeddings) and select an appropriate threshold based on this assumption for each of the clients. For the high epistemic uncertainty points of the global model, a similar thresholding can be performed to optimize its prediction quality.

## 5 EXPERIMENTS

In this section, we assess the effectiveness of our proposed method through a series of thorough experiments. We employ seven diverse datasets: MNIST (LeCun et al., 1998b), Fashion-MNIST (Xiao et al., 2017), MedMNIST-A, MedMNIST-C, MedMNIST-S (Yang et al., 2021; 2023), CIFAR10 (Krizhevsky, 2009), and SVHN (Netzer et al., 2011). The LeNet-5 (LeCun et al., 1998a) encoder architecture is applied to the first five datasets, while ResNet-18 (He et al., 2016) is used for CIFAR10 and SVHN. Building on the ideas from (Charpentier et al., 2020; 2022), we implemented the Radial Flow (Rezende & Mohamed, 2015) normalizing flow due to its lightweight nature and inherent flexibility. In all our experiments, we focus on a heterogeneous data distribution across clients. We consider a scenario involving 20 clients, each possessing a random subset of 2 or 3 classes. However, the overall amount of data each client possesses is approximately equal. The federated learning process is conducted using `FedAvg` algorithm.

In the following sections, we present different experiments that highlight the strengths of our approach. We should note that we don't have a dedicated experiment to illustrate how approach deals with local ambiguous data as the vision datasets we are considering have actually very few points of this type.

### 5.1 ASSESSING PERFORMANCE OF THE SWITCHING MODEL

In this section, we assess the performance of our method, where the prediction alternates between local and global models based on the uncertainty threshold.

For each dataset, we have three types of models. First, a global model is trained as a result of the federated procedure, following `FedAvg` procedure. Then, every client, after the federated procedure, retains the resulting encoder network while the classifier and flow are retrained from scratch using only local data. The third type of model, the "switching" model, alternates between the first two models based on the uncertainty threshold set for each client.

Additionally, for each client, we consider the following three types of data: data of the same classes used during training (InD), data of all other classes (OOD), and data from all classes (Mix). For each of these datasets and data splits, we compute the average prediction accuracy (client-wise). The results of this experiment are presented in Table 2.

In this experiment, we compare the performance (accuracy score) of different (personalized) federated learning algorithms: FedAvg (McMahan et al., 2017), FedRep (Collins et al., 2021), PerFedAvg (Fallah et al., 2020), FedBabu (Oh et al., 2022), FedPer (Arivazhagan et al., 2019), FedBN (Li et al., 2021), APFL (Deng et al., 2020).

We want to emphasize, that the state-of-the-art performance on the in-distribution data is not the ultimate goal of our paper, our method performs on par with other popular personalized FL algorithms. The remarkable thing about our approach is that it can be reliably used on any type of input data.

For our method, FedPN, we used "switching" model. From the Table 2, we observe that our model's performance for InD data is typically comparable to the competitors.

For "Mix" data, our approach is the winner by a large margin. Note, that this data split is the most

| | FedAvg | | | FedRep | | | PerFedAvg | | | FedBabu | | | FedPer | | | FedBN | | | APFL | | | **FedPN** | | |
|---|---|---|---|---|---|---|---|---|---|---|---|---|---|---|---|---|---|---|---|---|---|---|---|---|
| Dataset | InD | OOD | Mix | InD | OOD | Mix | InD | OOD | Mix | InD | OOD | Mix | InD | OOD | Mix | InD | OOD | Mix | InD | OOD | Mix | InD | OOD | Mix |
| MNIST | 87.6 | 82.3 | 84.8 | 99.3 | 0.0 | 50.0 | 99.3 | 50.0 | 74.7 | 99.6 | 61.8 | 80.5 | 99.5 | 29.4 | 64.1 | 74.2 | 65.0 | 69.4 | 77.9 | 58.9 | 98.4 | 98.4 | 98.3 | **98.3** |
| FashionMNIST | 66.4 | 55.2 | 60.8 | 95.3 | 0.0 | 47.7 | 95.1 | 22.9 | 59.0 | 95.8 | 22.4 | 59.1 | 95.8 | 15.2 | 55.5 | 56.3 | 51.8 | 54.1 | 77.0 | 34.0 | 55.5 | 84.3 | 78.2 | **81.3** |
| MedMNIST-A | 58.1 | 47.5 | 52.3 | 96.9 | 0.0 | 48.5 | 96.2 | 12.5 | 55.4 | 97.3 | 8.1 | 53.4 | 97.7 | 12.6 | 56.2 | 47.8 | 43.3 | 45.5 | 98.0 | 49.0 | 74.4 | 96.2 | 94.9 | **95.5** |
| MedMNIST-C | 49.5 | 45.5 | 43.6 | 93.0 | 0.0 | 46.5 | 91.7 | 2.4 | 47.5 | 95.0 | 13.6 | 54.1 | 95.2 | 10.9 | 54.0 | 50.0 | 43.3 | 44.3 | 95.4 | 53.3 | 73.3 | 94.4 | 88.7 | **91.2** |
| MedMNIST-S | 38.9 | 34.8 | 33.0 | 87.4 | 0.0 | 43.8 | 86.7 | 3.2 | 45.8 | 90.5 | 5.7 | 48.0 | 90.9 | 6.0 | 48.5 | 40.5 | 38.9 | 37.2 | 91.8 | 34.0 | 61.8 | 86.9 | 75.5 | **80.0** |
| CIFAR10 | 27.6 | 23.3 | 25.6 | 81.2 | 0.0 | 40.6 | 73.3 | 0.0 | 36.8 | 84.1 | 1.4 | 42.7 | 84.1 | 0.6 | 42.4 | 35.2 | 28.6 | 32.0 | 62.4 | 15.0 | 38.7 | 59.1 | 28.8 | **44.1** |
| SVHN | 80.6 | 76.7 | 78.3 | 94.7 | 0.0 | 47.3 | 93.4 | 9.5 | 51.4 | 94.9 | 11.2 | 53.0 | 95.4 | 6.5 | 50.9 | 74.4 | 71.9 | 73.3 | 80.5 | 42.5 | 59.3 | 87.1 | 62.2 | **73.4** |

Table 2: In the table, we report accuracy scores, obtained by different algorithms using different data splits. InD means that we use in-distribution data of all the clients and corresponding local models. OOD means that we use local models of different clients, but evaluate it on the classes not presented in the corresponding train splits. Mix means a random mixture of InD and OOD, where the share of InD is 50% and the same for OOD. Values in **bold** – best results on mixed data.

| | Local | FedAvg | FedRep | PerFedAvg | FedBabu | FedPer | FedBN | APFL | **FedPN** |
|---|---|---|---|---|---|---|---|---|---|
| MNIST | 99.1 | 87.6 | 99.3 | 99.3 | 99.6 | 99.5 | 74.2 | 77.9 | 99.4 |
| FashionMNIST | 95.3 | 66.4 | 95.3 | 95.1 | 95.8 | 95.8 | 56.3 | 77.0 | 95.7 |
| MedMNIST-A | 96.2 | 58.1 | 96.9 | 96.2 | 97.3 | 97.7 | 47.8 | 98.0 | 99.0 |
| MedMNIST-C | 93.3 | 49.5 | 93.0 | 91.7 | 95.0 | 95.2 | 50.0 | 95.4 | 96.6 |
| MedMNIST-S | 87.8 | 38.9 | 87.4 | 86.7 | 90.5 | 90.9 | 40.5 | 91.8 | 90.7 |
| CIFAR10 | 77.6 | 27.6 | 81.2 | 73.3 | 84.1 | 84.1 | 35.2 | 62.4 | 75.1 |
| SVHN | 91.2 | 80.6 | 94.7 | 93.4 | 94.9 | 95.4 | 74.4 | 80.5 | 92.2 |

Table 3: In this table we report the accuracy of the in-distribution data, using only local models. We can see, that for our approach if we know that the input data comes from in-distribution, we can safely use a local models (without switching) and the results will be on-par with others. Underlined values are the best for a given dataset.

realistic practical scenario — all the clients aim to collaboratively solve the same problem, given different local data. However, due to the heterogeneous nature of the between-clients data distribution (covariate shift), local models cannot learn the entire data manifold. Therefore, occasionally referring to global knowledge is beneficial while still preserving personalization when the local model is confident.

Note, that for CIFAR10 all the methods are not working well. For our method, it means that either the learned density model does not distinguish well for in- and out-of distribution data, or the threshold was not chosen accurately.

In Table 3 we present results for our model on InD data, when only **local** models were applied (so no "switching" is used, compared to Table 2). This might be useful in scenarios when we have knowledge that data comes from the distribution of the local data. In this case, there is no sense to switch to global model. We see, that despite it was not the purpose of the work, our approach performs on par with other competitors, slightly outperforming them on some datasets.

# 6 CONCLUSION

In this paper, we proposed a personalized federated learning framework that leverages both a globally trained federated model and personalized local models to make final predictions. The selection between these models is based on the confidence in the prediction.

Our empirical evaluation demonstrated that, under realistic scenario, this approach outperforms both local and global models when used independently. While the model's capacity to handle out-of-distribution data is not perfect and depends on various factors, such as the quality of the global model and the selection of the threshold, our "switching" approach ultimately leads to improved performance. It also enhances the reliability of AI applications, underscoring the potential of our methodology in a broader context of federated learning environments.

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
