# OpenReview forum: "Dirichlet-based Uncertainty Quantification for Personalized Federated Learning with Improved Posterior Networks"
_ICLR.cc/2024/Conference — Submitted to ICLR 2024_

### Official Review · Reviewer_p9yL · 2023-10-29

**Soundness:** 3 good
**Presentation:** 3 good
**Contribution:** 2 fair
**Rating:** 5
**Confidence:** 4

**Summary:**

Within the realm of personalized federated learning, personalized models may exhibit suboptimal performance when confronted with data from disparate domains. This paper aims to devise a criterion rooted in uncertainty assessment, serving as a means to decide whether to employ a global model or a local model. This approach is envisioned to enhance performance on both in-distribution and out-of-distribution data.

**Strengths:**

The writing is of high quality.
The algorithm is underpinned by a solid theoretical foundation. They introduce a stop gradient operation subsequent to the analysis of the problems associated with off-the-shelf loss functions.

**Weaknesses:**

1. Novelty is limited

Seems that most of the framework is directly taken from previous works, and this paper just combines and applies these in the federated learning setting as an application. For example, the Dirichlet-based model formulation is the same as [1] and [2], and the criteria are based on [3] and [4]. The only modification seems to be the stop-gradient operation based on the analysis on page 6~7.

2. Cannot scale to more complex datasets like CIFAR-10

From Table 2, it seems that there is a significant drop in the in-distribution data. An improvement from $10+%$ to $20+%$ cannot mitigate the performance drop from $70+%$ to $50+$. This challenges the model ability in this Dirichlet-based framework, especially for even more complex datasets like CIFAR-100 and Tiny-ImageNet.

3. Additional computation

The authors mention that they have to use a density model to decide if they should use a global or local model. This extra step adds more work for the computer. Even if they use a simple model, the reason why the model doesn't perform well on hard datasets might be that this density model is not very strong. But making this special model stronger would require more computation.

[1] Bertrand Charpentier, Daniel Zügner, and Stephan Günnemann. Posterior network: Uncertainty estimation without ood samples via density-based pseudo-counts. Advances in Neural Information Processing Systems, 33:1356–1367, 2020.

[2] Bertrand Charpentier, Chenxiang Zhang, and Stephan Günnemann. Training, architecture, and prior for deterministic uncertainty methods. In ICLR 2023 Workshop on Pitfalls of limited data and computation for Trustworthy ML, 2023.

[3] Andrey Malinin and Mark Gales. Predictive uncertainty estimation via prior networks. Advances in neural information processing systems, 31, 2018.

[4] Alex Kendall and Yarin Gal. What uncertainties do we need in bayesian deep learning for computer vision? Advances in neural information processing systems, 30, 2017.

**Questions:**

Are there any explanations on why you can use entropy and average entropy as the criteria for Epistemic uncertainty and Aleatoric uncertainty respectively in Section 3.2?

---

> ### Author Response · Authors · 2023-11-16
>
> We thank the reviewer for their thorough review and feedback. We address their questions below.
>
> **1. Novelty is limited. Most of the framework is directly taken from previous works. Dirichlet-based model formulation is the same as [1] and [2], and the criteria are based on [3] and [4]. The only modification seems to be the stop-gradient operation based on the analysis on page 6~7.**
>
> We kindly disagree with this assessment. Our main innovation is suggesting a new approach to switching between local and global models based on uncertainty quantification.  We believe that this methodological novelty is the main contribution of our paper. Everything else is already an implementation of a general idea, and we agree that we mostly use techniques that were known previously. However, the problem with the loss function identified by us was not known before despite this loss function has been used in a series of high impact papers. Thus, we think that it is another non-trivial contribution of our paper.
>
>
> **2. Cannot scale to more complex datasets like CIFAR-10.  Significant drop in the in-distribution data in Table 2… An improvement from 10+ to 20+ cannot mitigate the performance drop from 70+ to 50+. This challenges the model ability in this Dirichlet-based framework, especially for even more complex datasets like CIFAR-100 and Tiny-ImageNet.**
>
>
> We want to emphasize that it is not the problem of the particular method. All the datasets dropped in quality on this dataset for a given specific data heterogeneity. Please note that our method still improved upon the others.
>
>
> **3. Additional computation. Learning a density model adds more work for the computer. Even for a simple model, the reason why the model doesn't perform well on hard datasets might be that this density model is not very strong. But making this special model stronger would require more computation.**
>
>
> We agree with this point. Learning an additional density model is indeed of extra cost and may be considered as the limitation of the model.
> However, we should emphasize that not only the density model is important, but also the feature extractor itself. If the resulting features are good, then we don’t need to train a stronger density model given the intuition that representations are well clustered.
> Also, sometimes simple density models work pretty well [1, 2].
>
> Additionally, adding a density model is not just for the sake of improving the performance. It allows us to receive good uncertainty estimates and distinguish between different types of it. So the extra price is justified.
>
>
> [1] Bertrand Charpentier, Oliver Borchert, Daniel Zügner, Simon Geisler, and Stephan Günnemann. Natural posterior network: Deep bayesian predictive uncertainty for exponential family distributions. In International Conference on Learning Representations, 2022.
>
> [2] Vazhentsev, Artem, et al. "Efficient Out-of-Domain Detection for Sequence to Sequence Models." Findings of the Association for Computational Linguistics: ACL 2023. 2023.
>
>
> **Q1: Are there any explanations on why you can use entropy and average entropy as the criteria for Epistemic uncertainty and Aleatoric uncertainty respectively in Section 3.2?**
>
>
> The idea of using the entropy of the predicted Dirichlet distribution is rooted in the paper of [1, 2]. There is a connection between the sampling from the posterior of weights and the simplex, parameterized by Dirichlet. Specifically, Bayesian Model Averaging involves sampling from the posterior over weights, resulting in the posterior predictive distribution. This sampling from the posterior is an implicit sampling from the simplex, and we can parameterize the distribution over this simplex directly. The spread over the simplex is an indicator of the disagreement between models and thus can be used as an estimate of epistemic uncertainty. This spread can be measured, for example, as the entropy of the predictive Dirichlet.
> About aleatoric uncertainty - we follow ideas from [3]. Since aleatoric uncertainty is well-defined only for in-distribution samples, then every sample of the weights from posterior over weights should result in some ambiguous distribution over the simplex. That is why we find entropy first (which will be big for ambiguous samples), and then average it over all possible samples. Thus we result in the expected entropy as in equation (4).
>
>
> [1] Malinin, A., & Gales, M. (2018). Predictive uncertainty estimation via prior networks. Advances in neural information processing systems, 31.
>
> [2] Malinin, A., & Gales, M. (2019). Reverse kl-divergence training of prior networks: Improved uncertainty and adversarial robustness. Advances in Neural Information Processing Systems, 32.
>
> [3] Kendall, A., & Gal, Y. (2017). What uncertainties do we need in bayesian deep learning for computer vision?. Advances in neural information processing systems, 30.

---

> ### Comment · Reviewer_p9yL · 2023-11-22
>
> Thanks the authors for the rebuttal. Regarding the experimental results, I acknowledge that FedPN outperforms the baselines in smaller datasets, but I observe a more pronounced trade-off between InD and OOD performance in FedPN on CIFAR-10. Specifically, the improvement in OOD performance from 0 to 28% doesn't seem to fully compensate for the decline from 80% to 60% in comparison to the baselines. This raises my curiosity about FedPN's effectiveness on more complex datasets like CIFAR-100 and Tiny-ImageNet.
>
> In terms of novelty, while I recognize some new analysis in your work, I find it to be somewhat incremental.
>
> As such, I am inclined to maintain my original score.

---

> > ### Author Response · Authors · 2023-11-23
> >
> > Dear Reviewer,
> >
> > Thank you for the feedback.
> >
> > In response to your comments, we have carried out additional experiments and included these in the revised manuscript. We also invite you to review our common response addressed to all reviewers for a comprehensive overview of the changes made.
> >
> > Concerning the CIFAR-100 experiments, we have integrated them within our existing setup and observed that our model scales effectively to this dataset. However, to achieve optimal results, it has become apparent that an increase in the capacity of the flow model is necessary. We are working on this enhancement and are confident in presenting the finalized experiment in time for the Camera-Ready submission, should our paper be accepted.
> >
> > We appreciate your feedback and hope that our revisions meet your expectations.

---

### Official Review · Reviewer_g6z9 · 2023-10-30

**Soundness:** 2 fair
**Presentation:** 3 good
**Contribution:** 3 good
**Rating:** 3
**Confidence:** 4

**Summary:**

The paper sets out to introduce uncertainty quantification to Federated Learning. This can be particularly useful in deciding whether to use a local or global (federated) model for each user and sample. To this end, the paper uses a deep Bayesian algorithm, NatPN, with a slight modification to the loss function. Experiments show that the proposed framework can reduce the error on the OOD data while (often) not degrading on the InD data compared to existing baselines in Federated Learning.

**Strengths:**

- Judging by the Introduction, the research question is relevant to Federated Learning, and the contributions are original.
- The conceptual scheme in 2.1 makes sense. The paper clearly explains the different kinds of uncertainties and their importance in the context of Federated Learning.
- Design choices such as Dirichlet-based models, pseudo-count representation of $\alpha$, and entropy-based uncertainty estimates, seem sensible.
- The core experimental setup, i.e., dividing the data into InD and OOD, is sound.

**Weaknesses:**

- The methodological novelty is limited to applying existing approaches to a new problem.
- The authors are right to note that the choice of the uncertainty threshold is the most subjective part of the approach. Ablations (where the threshold is varied) are required to inspect how important this choice is. This also leads me to a concern about the conceptual scheme described in 2.1: a misestimated threshold may result in abstaining from predictions in a large portion of the data. Perhaps, the global model should not abstain at all?
- I am not sure about best practices in image classification these days, but the datasets seem simple.
- One issue I have with Table 2 is that the Mix score depends on the mixing proportions. I think it would be more illustrative to divide this table into InD and OOD tables and highlight with bold the best models in the respective tables.
- Ablations are required to complement the fix proposed in 3.3. Otherwise, I remain unconvinced that the described issue “could be a potential issue in federated learning” or even exists.

Minor issues
- Some parts were unclear from the submission and I had to read cited sources to understand them. After eq. 2, properties of p(g(x)) are listed, but it isn’t explained why f(g(x)) is required (as far as I understood, this term is a class prior, but then I do not exactly understand why it is required along with $\alpha_{prior}$). Eq. 5 is hard to parse as cross-entropy, consider using $CE$ like Charpentier et al. (2020).

**Questions:**

- Can there be scenarios where the scheme in 2.1 should be reversed? I.e., where the global model may have higher epistemic uncertainty.
- Could the general approach be extended to other Federated tasks such as Regression or RL?
- From the bounds of $\psi(x)$ before eq. 7, wouldn’t it make more sense to approximate it as $log(x) - 1/x$? Would it change the conclusions about issues with the loss function?

---

> ### Author Response · Authors · 2023-11-16
>
> We thank the reviewer for their thorough review and feedback. We address their questions below.
>
> **1. The methodological novelty is limited to applying existing approaches to a new problem.**
>
>
> We kindly disagree with this assessment. We are not considering any new problem but rather suggest a new approach to switching between local and global models based on uncertainty quantification.  We believe that this methodological novelty is the main contribution of our paper. Everything else is already an implementation of a general idea, and we agree that we mostly use techniques that were known previously. However, the problem with the loss function identified by us was not known before despite this loss function has been used in a series of high impact papers. Thus, we think that it is another non-trivial contribution of our paper.
>
>
> **2. Ablation for the threshold selection.**
>
> This is an interesting and important point, and we are currently working on the corresponding experiment. As soon as it is ready, we will add it to the text.
>
> - This also leads me to a concern about the conceptual scheme described in 2.1: a misestimated threshold may result in abstaining from predictions in a large portion of the data. Perhaps, the global model should not abstain at all?
>
> This is true, that misestimated threshold may lead to the abstaining for a large portion of the data. However, the models (local and global) have different thresholds. One can set the threshold for the global moel high enough so that it rarely abstains. However, the applications that imply the possibility of abstention of any predcition (for example, when it possible to refer to human expert) may benefit from the abstention of the global model.
>
>
> **3. I am not sure about best practices in image classification these days, but the datasets seem simple.**
>
> Thank you for your comment. We wish to emphasize that the datasets we employed are standard in the federated learning (FL) community and widely recognized as adequate for evaluating federated algorithms. Seminal works in FL, such as FedProx [1], SCAFFOLD [2], and FedOpt [3], have also utilized these datasets. We believe our evaluation effectively demonstrates the viability of our proposed approach.
>
> We acknowledge the reviewer's concerns regarding the complexity of the datasets. We will endeavor to include additional evaluations on a more complex dataset by the end of the discussion period or in the camera-ready version of our paper. However, we anticipate that the results will align with those already presented in our manuscript.
>
> [1] Li, T., Sahu, A. K., Zaheer, M., Sanjabi, M., Talwalkar, A., & Smith, V. (2020). Federated optimization in heterogeneous networks. Proceedings of Machine learning and systems, 2, 429-450.
>
> [2] Karimireddy, S. P., Kale, S., Mohri, M., Reddi, S., Stich, S., & Suresh, A. T. (2020, November). Scaffold: Stochastic controlled averaging for federated learning. In International conference on machine learning (pp. 5132-5143). PMLR.
>
> [3] Reddi, S., Charles, Z., Zaheer, M., Garrett, Z., Rush, K., Konečný, J., ... & McMahan, H. B. (2020). Adaptive federated optimization. ICLR 2021
>
>
>
> **4. One issue I have with Table 2 is that the Mix score depends on the mixing proportions. I think it would be more illustrative to divide this table into InD and OOD tables and highlight with bold the best models in the respective tables.**
>
> Thank you for the suggestion! We have split the table into several ones and added them to the supplementary part of the paper.
>
>
> **5. Ablations are required to complement the fix proposed in 3.3. Otherwise, I remain unconvinced that the described issue “could be a potential issue in federated learning” or even exists.**
>
> We apologize for being imprecise by saying that this led to issues in federated learning. In fact, the issue is general, and it holds not only for the federated scenario but also for the centralized one. Thank you for pointing it out!
> The demonstration of the issue is already in Figure 2. However, to make it even more apparent, we will add the same plot for the loss function with the proposed fix.
> Even further, we slightly change the locations of the point clouds. The reason for that is the untrained flow by default initializes higher density in the origin. So we move the noisy blob to the point of (0, 1) and add new results.

---

> > ### Author Response · Authors · 2023-11-16
> >
> > **6. Some parts were unclear from the submission… Why do we need f(g(x)) ? CE instead of eq.5**
> >
> > Following [1], we consider the update in the parameters of the prior distribution as softmax prediction, multiplied by the corresponding density. Doing this, we approximate the amount of “pseudo-counts”. This is an important component of this method, as the density part p(g(x)) estimates the whole evidence budget, and the classification part f(g(x)) distributes it among the classes.
> >
> > Regarding the equation with the cross-entropy. Note that as a part of the loss function we use so-called Uncertain Cross-Entropy, which is the expectation of the standard Cross-Entropy with respect to the predicted Dirichlet distribution. We agree that the form of writing might seem unusual and we will add a note to make the connection clearer.
> >
> >
> > [1] Bertrand Charpentier, Oliver Borchert, Daniel Zügner, Simon Geisler, and Stephan Günnemann. Natural posterior network: Deep bayesian predictive uncertainty for exponential family distributions. In International Conference on Learning Representations, 2022.
> >
> >
> > **Q1. Can there be scenarios where the scheme in 2.1 should be reversed? I.e., where the global model may have higher epistemic uncertainty.**
> >
> > We think that such situations are possible but in a circumstances that are much more special than the ones considered in our paper. For example, it might be that global model is very complex and there is not enough data for the confident prediction at some part of the input space. However, local model for some agents might have significantly simpler structure that might be confidently learned with fewer data.
> >
> >
> > **Q2. Could the general approach be extended to other Federated tasks such as Regression or RL?**
> >
> > For regression the answer is yes. Since the method for which we quantify the uncertainty is based on the NatPN [1], which works for any likelihoods from the exponential family of distributions, including Gaussian, we can easily apply it to regression as well. For RL it looks entirely possible as well.
> >
> > **Q3. Asymptotic log(x) - 1/x. Would it change analysis?**
> >
> > Thank you for the question! In fact, in our analysis, we assume that the model has sufficient evidence at the points, which lead to big alphas. Thus, the term 1/x can be omitted.
> > Regarding the validity of the result -- the experiment shown in Figure 2 computes the formula fairly. And we observe the described effect.
> >
> >
> >
> > [1] Bertrand Charpentier, Oliver Borchert, Daniel Zügner, Simon Geisler, and Stephan Günnemann. Natural posterior network: Deep bayesian predictive uncertainty for exponential family distributions. In International Conference on Learning Representations, 2022.

---

> > > ### Comment · Reviewer_g6z9 · 2023-11-16
> > > **Response to rebuttal**
> > >
> > > Thank you for the detailed response! It helps my understanding.
> > >
> > > 1. Yes, I think we mean the same thing. I didn't mean that the problem itself is new, but rather that applying this approach to this problem is new.
> > >
> > > 2. Let me know how it goes!
> > >
> > > 3. Could you please also provide references to some newer publications in FL? Other reviewers also seem to have problems with datasets.
> > >
> > > 5 and Q3. Ok, the issue is demonstrated to be real for the loss function. Still, this contribution seems too independent at the moment. If we removed this part from the text and used the old loss function, would the experimental results in section 5 change? Would the performance drop? An ablation is required to investigate.
> > >
> > > Q1 and Q2. Thank you for the clarifications, consider adding those to the text.

---

> ### Author Response · Authors · 2023-11-23
>
> Dear Reviewer,
>
> Thank you for your valuable feedback!
>
> Following your suggestions, we have conducted additional experiments, focusing on threshold selection ablation (detailed in Section B.4) and evaluating the effectiveness of the Stopgrad technique (described in Section B.5). We kindly invite you to review these additions, along with our general response to all reviewers.
> In relation to your query about recent publications, see for example these:
> 1) Yu, S., Hong, J., Wang, H., Wang, Z., & Zhou, J. (2022, September). Turning the curse of heterogeneity in federated learning into a blessing for out-of-distribution detection. In The Eleventh International Conference on Learning Representations
>
> 2) Kumar, S., Lakshminarayanan, A., Chang, K., Guretno, F., Mien, I. H., Kalpathy-Cramer, J., ... & Singh, P. (2022, September). Towards more efficient data valuation in healthcare federated learning using ensembling. In International Workshop on Distributed, Collaborative, and Federated Learning (pp. 119-129). Cham: Springer Nature Switzerland.
>
> 3) Li, B., Shi, Y., Kong, Q., Du, Q., & Lu, R. (2023). Incentive-Based Federated Learning for Digital Twin Driven Industrial Mobile Crowdsensing. IEEE Internet of Things Journal.
>
> 4) Zhai, R., Chen, X., Pei, L., & Ma, Z. (2023). A Federated Learning Framework against Data Poisoning Attacks on the Basis of the Genetic Algorithm. Electronics, 12(3), 560.

---

> ### Comment · Reviewer_g6z9 · 2023-11-30
> **Final response**
>
> Thank you for the update.
>
> Unfortunately, I am not sure that my concerns are addressed. I admit that I might not fully understand the results in B4, but it seems to me that we might as well use the global model if we care about OOD data, given that 1) using the local model in any capacity hurts OOD performance (i.e., there is always a trade-off between InD and OOD), and 2) selecting the threshold incorrectly may ruin the performance OOD while tuning it requires access to OOD data.
>
> Thank you for providing B5. I suggest using a log scale for the y-axis.
>
> Reading discussions with the other reviewers, it seems like there is still ongoing work on experiments with harder datasets. This is great and I wish the authors success in this endeavor.

---

### Official Review · Reviewer_LHCT · 2023-10-31

**Soundness:** 3 good
**Presentation:** 2 fair
**Contribution:** 3 good
**Rating:** 3
**Confidence:** 5

**Summary:**

Authors propose to use Dirichlet models for uncertainty quantification for personalized federated learning setup.

**Strengths:**

(1) First time using Dirichlet methods to assess uncertainty quantification in PFL to deal with OOD and mix of In & OOD.

(2) I liked the idea of showing Mix and OOD data in experiments. Especially the Table 2. I have more comments in Weaknesses though.

**Weaknesses:**

(1) Authors claim to use Dirichlet methods for uncertainty quantification but do not compare with existing Bayesian federated methods such as FedPA[1], FedEP[2] , and FedPop[3], which are also assumed to perform efficient Bayesian inference for (P)FL.


(2) Comparison is unfair for multiple reasons : 1)  Other models do not use a switching model. Technically speaking, other personalized FL models can benefit from "switching" too, but not exactly with the same way as this paper does. Also, the proposed method is the only method geared toward uncertainty quantification.  Bayesian FL methods and/or the methods mentioned above should be included as those methods are  (in)directly uncertainty quantification methods too. The paper claims that Dirichlet is an efficient method compared to MCMC and VI, but what about in terms of performance in federated settings?

(3) 20 clients is practically low, and having an equal number of data points is not a practical assumption. I think 100 clients and using more difficult datasets such as CIFAR100 and Stackoverflow would be needed.  Also, an ablation study over the number of clients would be needed, too.

(4) The claim "In this paper, we proposed a personalized federated learning framework that leverages both a globally trained federated model and personalized local models to make final predictions." is misleading as there are previous papers utilizing both global and local models for prediction [4]

(5) The transition from Section 3 to federated settings is poorly developed. Section 3 has too much text in it it is not clear what are the problems/complications of using Dirichlet methods in federated settings. Authors should condense Section 3 and move some parts to the Appendix as it is hard to understand what is the problem for federated learning.


[1] Al-Shedivat, Maruan, et al. "Federated learning via posterior averaging: A new perspective and practical algorithms." ICLR'21 arXiv:2010.05273 (2020).
[2] Guo, Han, et al. "Federated Learning as Variational Inference: A Scalable Expectation Propagation Approach." ICLR '23arXiv:2302.04228 (2023).
[3] Kotelevskii, Nikita, et al. "Fedpop: A bayesian approach for personalised federated learning." Advances in Neural Information Processing Systems 35 (2022): 8687-8701.
[4] Hanzely, Filip, and Peter Richtárik. "Federated learning of a mixture of global and local models." arXiv preprint arXiv:2002.05516 (2020).

**Questions:**

(1) Why is there no comparison with previous Bayesian FL methods such as such as FedPA[1], FedEP[2], and FedPop[3] ?

(2) Federated loss was unclear to me. Are you using the loss in Eqn. (8)?



[1] Al-Shedivat, Maruan, et al. "Federated learning via posterior averaging: A new perspective and practical algorithms." ICLR'21 arXiv:2010.05273 (2020).
[2] Guo, Han, et al. "Federated Learning as Variational Inference: A Scalable Expectation Propagation Approach." ICLR '23arXiv:2302.04228 (2023).
[3] Kotelevskii, Nikita, et al. "Fedpop: A bayesian approach for personalised federated learning." Advances in Neural Information Processing Systems 35 (2022): 8687-8701.

---

> ### Author Response · Authors · 2023-11-16
>
> We thank the reviewer for their thorough review and feedback. We address their questions below.
>
> 1. **Comparison with Bayesian Baselines FedPA[1], FedEP[2], and FedPop[3]:**
>
> In the FedPA paper, the authors adopt a Bayesian approach to derive a global posterior from locally distributed data. This focus leads to an optimization process that results in the global model, lacking personalization. The adaptation of our switching procedure to their model is not straightforward. Similarly, the FedEP paper also focuses on the global posterior without offering personalized models, making it unclear how to compare our local/global models with their method.
>
> As for the FedPop paper, while it adopts a Bayesian view and has personalization, they don’t have a notion of a global model. They indeed have a shared backbone (feature extractor) and many local personalized models. However, no proper global model. Moreover, they do not provide a method for separately evaluating epistemic and aleatoric uncertainties.
>
> We wish to highlight that our paper's *main contribution is the introduction of the switching framework.* This framework is general, allowing for various implementations, whether they are based on Variational Inference (VI) or Markov Chain Monte Carlo (MCMC) methods. We chose to illustrate this framework with a Dirichlet-based model because of its relative ease of training, straightforward implementation, and intuitive way of measuring epistemic uncertainty.
>
>
> **2. Comparison is unfair for multiple reasons:**
>
> - Other models do not use a switching model
>
> We appreciate the observation regarding the lack of switching strategies in other methods. Indeed, our approach is unique in this aspect. Implementing a similar switching strategy in other models is not a straightforward task.
>
> This complexity arises from the complications involved in identifying a suitable metric for switching that aligns with the specific characteristics and objectives of each model. Our method was specially designed with a switching strategy that is integrated into its core framework, making the replication of this feature in other models non-trivial. This feature not only demonstrates the innovation of our approach but also underscores the challenges in adapting such strategies to different models. Moreover, not all the models used in the list of baselines we compared with are personalized. Hence, it is questionable what are those models between which we should switch.
>
> - The proposed method is the only method geared toward UQ. Bayesian FL methods and/or the methods mentioned above should be included as those methods are (in)directly uncertainty quantification methods too.
>
> Uncertainty quantification is an essential part of our approach that gives the full UQ capabilities for the resulting system of models. However, our main innovation is not in UQ itself but rather in its application to switching between local and global models. That is why our experiments are focused on classical performance metrics such as accuracy, and not on the evaluation of predictive uncertainty. Even if we have an estimate of uncertainty for some model, it is not clear how we should proceed with them for switching as some of the mentioned models do not have a global model at all. For example, in FedPop there is no global model, but only local ones (which are compositions of the global parameters and local ones sampled from some population prior).
>
> - The paper claims that Dirichlet is an efficient method compared to MCMC and VI, but what about in terms of performance in federated settings?
>
> Indeed, our approach utilizing the Dirichlet method can be seen as a lightweight form of VI, offering greater computational efficiency compared to standard VI techniques and MCMC. The latter, particularly, requires significant time both to converge to the target posterior and to obtain uncorrelated samples.
> Our paper's primary contribution is the introduction of a novel switching framework. We specifically chose the Dirichlet-based model within this framework for its computational lightness and its advantages in practical scenarios. While it's true that employing MCMC with many computational resources might yield superior results, our aim is not to demonstrate state-of-the-art performance. Rather, we focus on presenting a new concept in model selection that was previously unexplored, along with validating its effectiveness through practical application.

---

> > ### Author Response · Authors · 2023-11-16
> >
> > **3. 20 clients is practically low, and having an equal number of data points is not a practical assumption… Cifar 100 with 100 clients… Ablation study over the number of clients**
> >
> > We kindly disagree that 20 clients is a small number of clients, and there are a bunch of recently published papers that follow the same setup. However, we agree that adding more clients can improve the impression of the method, and we are working on the experiment for CIFAR100 with 100 clients. We will report the result as soon as it is ready.
> > Regarding the different number of data points, it is a good point and definitely could be one of the next steps. However, we stress that it is not the weakness of the paper. In our paper, we study the particular setup in which we focus on the heterogeneity of data (covariates). One can also consider the heterogeneity in terms of number of points per client. However, it is another type of heterogeneity. This will add an extra level of complexity, and could complicate the analysis.
> > We will discuss it as a limitation explicitly.
> >
> >
> > **4. The claim "In this paper, we proposed a personalized federated learning framework that leverages both a globally trained federated model and personalized local models to make final predictions." is misleading as there are previous papers utilizing both global and local models for prediction [4]**
> >
> > Thank you for your comment and for referring to the work of Hanzely & Richtárik. We acknowledge this prior work and have cited it in our manuscript. We would like to clarify that our claim is not about being the first to consider both local and global models. Rather, our contribution lies in proposing a framework that utilizes these models differently, specifically by switching between purely local and global models based on uncertainty quantification.
> >
> > The focus of Hanzely & Richtárik's work differs significantly from ours. They employ the average distance of local models to their collective mean as a regularization technique, linking local models. Conversely, our approach relies on either purely local training or the global model, depending on model uncertainty for each individual sample. Additionally, the "averaged model" in their research is not akin to a global model in design; it does not guarantee universal performance across all data and is more similar to the MAML approach.
> >
> > A key distinction between our work and the model by Hanzely & Richtárik is our use of the switching mixture of global and local models for inference, whereas they provide a transition between these models in their objective through a regularization penalty $\lambda$. In their model, there is no switching for inference: a single model is used to evaluate all local samples, with $\lambda$ serving as a hyperparameter. Here, $\lambda = 0$ corresponds to purely local models, and $\lambda = \infty$ to a global model. Lastly, their model is not Bayesian, unlike ours.
> >
> > We hope this clarifies our framework's unique aspects and contributions in the context of existing literature.
> >
> >
> > **5. transition from Section 3 to federated settings is poorly developed. Section 3 has too much text in it it is not clear what are the problems/complications of using Dirichlet methods in federated settings. Authors should condense Section 3 and move some parts to the Appendix as it is hard to understand what is the problem for federated learning.**
> >
> >
> > We would like to thank the reviewer for this important comment. We certainly agree that currently we do not have enough focus on the specific issues of FL and have some extra details that can be easily moved to the Appendix. We think that these issues are easy to resolve and we will make sure to improve the readability for the camera-ready version of the paper (if accepted).
> >
> >
> > **Q1 Why is there no comparison with previous Bayesian FL methods such as such as FedPA[1], FedEP[2], and FedPop[3] ?**
> >
> > We address the question in our answer to the Weakness 1.
> >
> >
> > **Q2: Federated loss was unclear to me. Are you using the loss in Eqn. (8)?**
> >
> > Yes, we use the loss function from equation (8) for training our model.
> > Let’s break this loss into parts. The first one is the UCE (uncertain cross entropy) [1], which is the expectation of CE loss with respect to the predicted Dirichlet distribution, but without gradient update with respect to the parameters of a density model. This part pushes the model to make correct predictions and be confident. The second term penalizes predicted Dirichlet for overconfident predictions and is the same the one in [1]. The last term is the correction we suggested.
> >
> > [1] Bertrand Charpentier, Oliver Borchert, Daniel Zügner, Simon Geisler, and Stephan Günnemann. Natural posterior network: Deep bayesian predictive uncertainty for exponential family distributions. In International Conference on Learning Representations, 2022.

---

> > > ### Comment · Reviewer_LHCT · 2023-11-23
> > > **Response to Authors**
> > >
> > > Thanks to the authors for the rebuttal. I understand that the contribution is switching model, but the main problem is the evaluation of and ablation over "switching model."
> > >
> > > I am inclined to maintain my original score.

---

> > > > ### Author Response · Authors · 2023-11-23
> > > > **Comment by Authors**
> > > >
> > > > Dear reviewer,
> > > >
> > > > we are not sure that we fully understand the depth of the issue with evaluation. For example last year ICLR accepted 41 papers that mention federated in their title, out of those 8 do not contain evaluation on image classification problems. For the rest 33 papers, there are only 4 that consider more complex evaluation than CIFAR, namely Tiny-ImageNet (3), ImageNet (1). Also, there is one paper with no experiments and two with only logistic regression for evaluation among those 33. Thus, we believe that our work is completely in-line with community accepted level of evaluation for federated learning papers.
> > > >
> > > > Regarding the ablations, following your suggestions, we have conducted additional experiments, focusing on threshold selection ablation (detailed in Section B.4) and evaluating the effectiveness of the Stopgrad technique (described in Section B.5).
> > > >
> > > > We kindly ask you to look again on our results and to acknowledge our evaluation and ablation studies.

---

### Meta-Review · Area_Chair_n11Q · 2023-12-05

**Metareview:**

This paper presents a method for personalized federated learning (FL) with heterogeneous data across clients. To handle OOD data, the proposed method uses a mix of local and global models and depending on whether an input is in-distribution or OOD, it selects the appropriate model based on its confidence. It uses the idea of posterior networks to quantify uncertainty.

While the reviewers appreciate the basic idea, there were several concerns regarding, some of which include (1) existence of other Bayesian FL methods that could be applicable in such settings with some modifications and it isn't clear the specific benefits the posterior network based formulation would bring unless there is some comparison with such methods, (2) the switching idea that is the key here can also be applied with other personalized FL methods and the paper should consider comparing with them, (3) the experiments used a fairly small number of clients, and (4) there were concerns on when to choose the global vs local models.

The authors submitted their response; however, the concerns remained, and in the end, it is felt that the paper is not yet ready to be published. The authors are advised to address the concerns raised in the reviews and consider submitting the paper at another venue.

**Justification For Why Not Higher Score:**

The paper had several issues as pointed out in the meta review

**Justification For Why Not Lower Score:**

N/A

---

### Decision · Program_Chairs · 2024-01-16

Reject